# Massaranduba Sawdust: A Potential Source of Charcoal and Activated Carbon

**DOI:** 10.3390/polym11081276

**Published:** 2019-07-31

**Authors:** Jonnys P. Castro, João Rodrigo C. Nobre, Alfredo Napoli, Maria Lucia Bianchi, Jordão C. Moulin, Bor-Sen Chiou, Tina G. Williams, Delilah F. Wood, Roberto J. Avena-Bustillos, William J. Orts, Gustavo H. D. Tonoli

**Affiliations:** 1Department of Forest Products (DPF), Forest Institute (IF), Federal Rural University of Rio de Janeiro, Rodovia BR 465, Km 07, C.P. 74527, Seropédica 23890-000, Brazil; 2Department of Wood Technology (DETM), State University of Pará, Campus VI, Rodovia PA-125, Angelim, Paragominas 68625-000, Brazil; 3Biomass, Wood, Energy, Bioproducts Research Unit, CIRAD, 73 Rue Jean François Breton, CEDEX 5, 34398 Montpellier, France; 4Department of Chemistry (DQI), Federal University of Lavras, C.P. 3037, Lavras 37200-000, Brazil; 5Department of Forest Science (DCF), Federal University of Lavras, C.P. 3037, Lavras 37200-000, Brazil; 6Bioproducts Research, USDA ARS WRRC, Albany, CA 94710, USA; 7Healthy Processed Foods Research, USDA ARS WRRC, Albany, CA 94710, USA

**Keywords:** Amazonian woods, lignocellulosic, lignin, pyrolysis, CO_2_, surface area

## Abstract

This paper provides proof of concept that activated carbon (AC) may be readily produced using limited conversion methods and resources from sawdust of massaranduba (*Manilkara huberi*) wood, thereby obtaining value-added products. Sawdust was sieved and heat-treated in an oxygen-free muffle furnace at 500 °C to produce charcoal. The charcoal was activated in a tubular electric furnace at 850 °C while being purged with CO_2_ gas. Microstructural, thermal and physical properties of the three components: sawdust, charcoal and AC were compared by means of field emission scanning electron microscopy (FESEM), X-ray diffractometry (XRD), thermogravimetric analysis (TGA), differential scanning calorimetry (DSC), density and water adsorption/desorption measurements. The resulting AC had a large surface area as measured by Brunauer-Emmett-Teller (BET) comparable to other such values found in the literature. The large surface area was due to pore development at the microstructural level as shown by FESEM. XRD illustrated that sawdust had a semi-crystalline structure whereas charcoal and AC evidenced mostly amorphous structures. TGA and DSC showed that AC had high reactivity to moisture compared to sawdust and charcoal.

## 1. Introduction

Activated carbon (AC) is a renewable resource used in a variety of industrial applications and has potential use in many other applications [1]. Removal of impurities by AC is attributed to its large surface area and reactive functional groups with affinity for impurities [2]. AC filtration of water improves taste, eliminates pigments, and reduces UV absorbance, oxidation and odor [3]. Duration and quantity determine the effectiveness of interactions (i.e., removal of impurities) of adsorbents and impurities with AC treatment, i.e., more time and more AC will result in enhanced removal of impurities by adsorption [4].

High temperature treatments are required to produce activated carbon. Behazin et al. [5] produced biochar from soft wood chips and suggested that temperatures above 500 °C produced char with high surface area and porosity leading to high functionality. Das et al. [6] were able to increase the surface area of AC derived from pine sawdust from 28.7 m^2^·g^−1^ to 335.9 m^2^·g^−1^ by increasing the temperature from 350 °C to 900 °C, respectively, and showed that the dwell time of 1 h at maximum temperature was essential in successful activation. Das et al. [6] also heat-treated at 450 °C for 10 min, then treated with 70% ethanol, 10% hydrochloric acid followed by autoclaving, which we would not recommend as a standard treatment due to the toxicity of hydrochloric acid fumes upon opening the autoclave. The ethanol/acid treatment was then heated (500 °C for 1 h, giving a surface area of 255 m^2^·g^−1^) and proved to be less effective than heat-treatment alone (900 °C for 1 h, giving a specific surface area of 335 m^2^·g^−1^). Table 1 summarizes activation conditions, specific surface areas and micropore volumes of ACs derived from different precursors in the literature.

Low cost and high capacity adsorbents that can be used to treat different types of waste materials have been intensely investigated. For example, waste timber from sawmills may be converted on-site to biochar and used for remediation of sulfide tailings and filtration of water from mine wastes [13].

Solar reactors may be used to power pyrolysis. Zeng et al. [14] were able to achieve temperatures up to 2000 °C with controlled heating rates of up to 450 °C·s^−1^. Solar reactors supply a source of heat that increases the power generated from the raw material. Solar reactors reduce or eliminate contamination of the gaseous byproducts, reduce emissions and decrease the need for energy-intensive processing of pure oxygen [15].

AC has a carbonaceous, porous structure with heteroatoms, notably oxygen, that add functional groups. AC made from piassava (fiber palm, broom straw residue) had a highly developed porous structure with surface areas of 658 to 1190 m^2^·g^−1^ [2]. AC from piassava was activated by high heat and chemical additives in the absence of oxygen (under N_2_ flow). Large surface areas in AC provide numerous adsorption locations for impurities from both liquid and gas phases.

AC is classified into three types: granular, fibrous or powder; according to the particle size and shape of pores, with each type having a specific application [16]. AC produced from such materials as wood, bone, coconut husk and endocarp, sugarcane bagasse, fruit seeds and other waste biomass feedstocks have been studied [17] suggesting that all carbonaceous feedstocks have potential to produce AC.

Microstructural changes in biomass have shown characteristic changes in tissue during different stages of char production using poplar wood as an example. A light microscope equipped with a hot stage and a chamber through which nitrogen gas flowed was set up to mimic conditions in a reactor [18]. The poplar tissue expanded perpendicular to the wood grain from room temperature to 300 °C, corresponding to torrefaction. Lignocellulosic tissue density increased from 350 °C to 500 °C evidenced by tissue darkening and collapse. As the temperature exceeded 500 °C, the tissue reached its maximum state of contraction, was completely carbonized and was dense and black or nearly black.

Phosphoric acid (H3PO4) combined with heat has been used as a carbon activation method [19,20]. Treatment with strong basic chemicals (NaOH, KOH, K_2_CO_3_) has also been used to produce activated carbon from biomass residues [21]. ZnCl_2_ is yet another activation compound that has been used [22,23,24]. Freitas et al. [25] used a two-step heating process followed by treatment with HCl and water washing. Tovar et al. [26] took advantage of the acid remaining in the residue following pectin extraction to then use the residue for making activated carbon. The issue with all the aforementioned methods is that harsh chemicals were used to produce AC that then needed to be washed to remove the chemicals and then neutralized. The processes produced liquid waste that was a source of pollution. In addition, because the AC was wet, it had to be dried before use requiring energy. The process described, in this work, furthers the research of Nobre et al. [27] as a two-step process requiring heating only without washing resulting in a more environmentally-friendly method of production of AC.

The novelty of the present work includes the use of residual biomass from the Amazon region and a simple process requiring heat to produce AC. The Amazon is one of the world’s major tropical timber producing regions of the world. Industrial wood processing in the Amazon is the main economic activity, therefore, producing a value-added product that uses the wood residue would provide economic advantages, contribute to sustainability and resource utilization. According to the IMAZON Institute, the average lumber yield in the Amazonian sawmills is 41%; 14.2 million m^3^ of trees are processed, resulting in roughly 5.8 million m^3^ of lumber. Thus, the resulting residue constitutes 59% (8.4 million m^3^) [28,29] of the harvest. *Manilkara huberi* (Ducke) Chevalier (massaranduba) is one of the main species used in the timber industry in Brazil and a single log of massaranduba (~2.90 m^3^) yields 45% (1.30 m^3^) usable lumber [29] and the remaining 55% is biomass residue. Roughly 18% of the residue (0.33 m^3^) is sawdust. Massaranduba wood residues are currently burned in combustion chambers for power generation. An alternative, value-added product would be AC. The knowledge of the AC production from tropical woods is scarce. This work serves to catalyze production and scale-up trials for AC from wood residues of the Amazon. The Amazon has a large diversity of lignocellulosic residues and use of agricultural and wood residues to produce AC is a novel, value-added process requiring the development of optimization parameters to improve the characteristics of the product.

The objective of this work was to produce value-added activated carbon (AC) from massaranduba sawdust to and evaluate the thermal and physical properties and microstructures of all components in the production of AC.

## 2. Materials and Methods

### 2.1. Materials

Massaranduba (*Manilkara huberi*) sawdust, a byproduct residue of the tropical lumber industry, was used as the starting material for our studies. The sawdust was obtained from SEMASA Industry Trade and Export Wood Ltd., located in the metropolitan region of Belém, Pará state, Brazil. The average contents of lignin, extractives, ash and holocellulose (cellulose + hemicelluloses) of the massaranduba wood on a dry weight basis were 34.7%, 7.3%, 0.3% and 56.6%, respectively [30].

### 2.2. Production of the Charcoal Precursor

Sawdust was sieved through a 60-mesh screen, placed in an oxygen-free muffle furnace and pyrolyzed by heating at a rate of 100 °C·h^−1^ to 500 °C. The sample was held for 30 min at 500 °C and then cooled by natural convection. The resulting product was the charcoal precursor (charcoal).

### 2.3. AC Production

Wood charcoal was heated (rate of 10 °C·min^−1^) and maintained at 850 °C for 1 h in a tubular electric furnace containing a cylindrical reactor with CO_2_ flow rate of 150 mL·min^−1^. The resulting AC was then cooled by natural convection.

### 2.4. Characterization of Particle Morphology

Sawdust, charcoal and AC samples were attached to stubs with double adhesive coated carbon tabs (Ted Pella, Inc., Redding, CA, USA). Subsequently, the samples were sputtered with gold-palladium in a Denton Desk II coating unit (Denton Vacuum, LLC, Moorestown, NJ, USA) and then viewed and photographed in a Hitachi S-4700 field emission scanning electron microscope (FESEM). The FESEM micrographs were captured at 2650 × 1920-pixel resolution.

### 2.5. X-ray Diffraction (XRD)

XRD patterns (in duplicate) were obtained for sawdust, charcoal and AC powders using an X-ray multipurpose diffractometer (Philips X’Pert MPD, PANalytical, Inc., Westborough, MA, USA). Scattered radiation (CuKα at 45 kV and 40 mA) was detected in the range of 2θ = 5–40°, at a scan rate of 2°·min^−1^.

### 2.6. Thermogravimetric Analysis (TGA)

Sawdust pyrolysis were analyzed by TGA (py-TGA) in a Rubotherm instrument (Magnetic suspension balance). Sawdust samples (about 500 mg dry wt basis) were heated in an alumina crucible at a heating rate of 1 °C·min^−1^ from 40 °C to 500 °C under N_2_ gas flow (30 mL·min^−1^) at atmospheric pressure. Weight loss and derivative weight loss were determined.

Sawdust, charcoal and AC samples combustion were analyzed by TGA (Co-TGA) in a Perkin Elmer Pyris 1 TGA instrument (PerkinElmer, Waltham, MA, USA). Samples (about 10 mg dry wt basis) were heated in a Pt crucible at a rate of 10 °C·min^−1^ from 25 °C to 700 °C and an air flow rate of 60 mL·min^−1^. Critical weight loss temperatures (T_onset_) were determined by the intersection of the extrapolated line from the beginning of the thermal event, with the curve tangent in the thermal degradation [31] from the onset points of the Co-TGA curves.

### 2.7. Differential Scanning Calorimetry (DSC)

DSC of combustion (Co-DSC) was used to estimate the energy changes that occurred during dehydration of the sawdust, charcoal and AC samples. In typical DSC curves of lignocellulosic materials, the endothermic peak appears between 30 °C and 150 °C [32,33]. In this study, the peak area was measured and was related to the energy needed for removing water from the sample (heat of dehydration). The quantity of heat needed to dehydrate a sample is a measure of water reactivity, since accessibility to water is proportional to the available bonding groups.

The heat of dehydration was measured using DSC (TA Instruments, DSC 2910, New Castle, DE, USA). Sawdust, charcoal and AC samples were maintained at ambient temperature (~25 °C) in a 60% relative humidity (RH) chamber containing KI and NH_4_NO_3_ solutions for 24 h before each test (except for the dry activated carbon). The samples were purged with N_2_ gas at a flow rate of 60 L·min^−1^. Approximately 8 mg of sample were placed into each DSC pan and heated from 30 °C to 200 °C at a rate of 10 °C·min^−1^. The heat of dehydration was estimated using the Universal Analysis 2000 software (TA Instruments, New Castle, DE, USA) that integrates the peak areas among ranges of temperatures to calculate the corresponding energy variations.

### 2.8. True Density

True density of the sawdust, charcoal and AC were determined to monitor the influence of the burning conditions on particle structure. Ten values of true density were determined for each particle sample using a gas (Helium) pycnometer (AccuPyc II 1340 Series Pycnometer, Micromeritics Instrument Corp., Norcross, CA, USA).

### 2.9. Surface Area and Porosity Structure

AC surface areas were measured by adsorption and desorption of N_2_ in Autosorb-1 (Quantachrome Instruments, Boynton Beach, FL, USA). Surface area (S_BET_) was determined from the adsorption isotherms using the Brunauer-Emmett-Teller (BET) multipoint method. The pore volume distribution was obtained from the isothermal adsorption/desorption of N_2_, using the density functional theory (DFT) to model the functional pore density [34].

### 2.10. Water Isotherm Measurements

A dynamic vapor absorption analyzer (DVS-1, Surface Measurement Systems, London, UK) was used to measure the water sorption isotherms at 25 °C. The samples were hydrated to specific RHs to equilibrium. Each sample was exposed to moisture from 0 to 98% RH for adsorption cycle, and afterwards from 98% to 0% RH for the desorption cycle.

## 3. Results and Discussion

### 3.1. Morphological Characteristics

Drastic changes in the surface morphology of charcoal after pyrolysis with CO_2_ activation were observed using FESEM (Figure 1, Figure 2 and Figure 3). Massaranduba sawdust and charcoal had relatively flat, smooth surfaces (Figure 1 and Figure 2). In contrast, AC showed diverse porous structures with micro and nanometer pores (Figure 3).

Figure 3f (arrows) shows nanoscale pores on the surface of the AC. The appearance of pores on AC surfaces is a result of removal of volatiles during pyrolysis and by the CO_2_ reaction, effectively unblocking any clogged pores. Porosity development may also indicate the trapping of CO_2_ in the pores [35].

Activation of charcoal unclogged pores that were filled with substances that condensed from the carbonization process. Fibers were also detached during activation and formed fissures and cracks thereby increasing the surface area and adsorption capacity. Couto et al. [36] also noted the growth in the number of meso-and macropores after activation with CO_2_. In addition, Ghouma et al. [11] found heterogeneous macropores ranging from 10 µm to less than 1 µm in diameter.

### 3.2. XRD of the Different Particles

Among the wood components, cellulose is crystalline, while hemicellulose and lignin are amorphous [37]. XRD patterns were used to determine the degree of crystallinity under different conditions [38]. As expected, the XRD patterns of charcoal and AC (Figure 4a) displayed a diffuse background curve lacking significant peak intensity from 2θ = 10° to 2θ = 40°, indicating the presence of strictly amorphous domains. During pyrolysis, cellulose is converted to carbon-rich amorphous charcoal, therefore, charcoal and AC had no crystalline peaks. The XRD diffractograms of the sawdust had an amorphous broad hump and crystalline peaks typical of semi-crystalline materials. Sawdust is composed of lignocellulosic materials and showed a peak at 2θ = 22.6°, assigned to the (002) plane of cellulose I. The two weaker diffraction peaks at 2θ = 14.8° and 2θ = 16.3° are assigned to (101) and (10-1) lattice planes of cellulose I [39]. Charcoal and AC are pyrolyzed particles and exhibited a diffuse background curve without significant peaks, indicative of strictly amorphous domains.

### 3.3. Thermogravimetry

Thermogravimetric analysis (py-TGA) of sawdust pyrolysis (Figure 4b) showed three thermal conversion/decomposition stages after drying step (100 °C). The first stage occurred between 225 °C and 300 °C, with weight loss of about 15% for sawdust, a second main degradation stage occurred between 300 °C and 370 °C and the third conversion is obtained between 370 and 500 °C. Moreover, the decomposition temperatures for charcoal and AC were much higher than that of sawdust during combustion (Co-TGA) (Figure 4c). Poletto et al. [38] reported that the thermal pyrolysis of cellulose occurred at 350 °C for wood under N_2_ as was evidenced by noticeable peak at the temperature that matches the maximum rate of decomposition, similar to the results in the present study (Figure 4b). The depolymerization of hemicelluloses occurred between 180 and 350 °C, and the random decomposition of glycosidic bonds of cellulose was between 275 and 370 °C [40]. Lignin and residual lignin decomposition occurred between 250 °C and 500 °C [41]. At about 340 °C the dominant peak of DTGA in the sawdust was evident (Figure 4b). The dominant peak is formed in the temperature range corresponding to loss of hemicellulose mass (225–325 °C), residual lignin (250–500 °C) and cellulose (305–375 °C) [42].

The high lignin content of massaranduba wood favors the production of AC, since lignin is more resistant to thermal degradation than cellulose or hemicellulose. Wood with high lignin contents has greater yield than those with low lignin contents. Lignin contributes to the fixed carbon content at the end of carbonization and activation. Lignin can also promote the formation of phenolic groups on AC surfaces. High holocellulose contents can directly influence surface chemistry and promote lower acidity and lower amounts of carboxyl groups in the AC. The hydroxyl groups in cellulose and hemicellulose decompose easily because of their high thermal instabilities compared to the stable aromatic compounds of lignin. According to Deng et al. [43], cellulose and hemicellulose decompose and dehydrate leading to micropore development, whereas aromatic rings in lignin lead to nonporous carbonaceous materials during pyrolysis.

Thermal combustion (Co-TGA) of sawdust occurs essentially in two stages and may be observed in the derived thermogravimetric analysis (Co-DTGA) curves (Figure 4c,d). However, due to the use of oxygen flow during the experiment, combustion of the sample occurred so cannot be a direct indication of what happens during oxygen-free pyrolysis. In the first stage of the thermal treatment, hemicellulose and cellulose were volatilized, after partial conversion of lignin to carbon involving homogeneous combustion of the volatiles. The last step was the auto-combustion of the solid charcoal fraction through heterogeneous combustion. It is well-known that there are differences between thermal conversions occurring in an oxygen vs. oxygen-free atmospheres.

Charcoal and ACs, having undergone combustion, showed an increase in the onset degradation temperature (T_onset_) (Figure 4c) compared to sawdust. The Co-TGA (Figure 4c) and Co-DTGA (Figure 4d) of charcoal and AC showed that thermal degradation occurred at higher values than sawdust, with T_onset_ and maximum degradation temperature of 457 °C and 525 °C for charcoal and 533 °C and 600 °C for AC, respectively (Table 1).

In comparison, sawdust started to degrade at 255 °C, whereas charcoal and AC started to degrade at approximately 363 °C and 410 °C, respectively. In charcoal, the presence of aromatic compounds resulted in higher stored energy than in sawdust. During the thermal conversion process, less energy was used to degrade sawdust thereby conserving energy in unsaturated carbon structures, such as the aromatic rings in lignin than in AC or charcoal [44].

The difference in thermal properties was due to charcoal and AC containing lignin-rich structures, resulting in initial degradation temperatures higher than those of sawdust. Charcoal consists mostly of lignin [41]. Charcoal combusts at about 500 °C, whereas AC combusts at about 600 °C. Charcoal had higher volatile contents than AC, because during charcoal combustion, volatiles were burned at 400–500 °C, and this solid-gas heterogeneous combustion led to the loss of mass. The combustion of AC occurred later (at a higher temperature) than that of charcoal because few volatiles remained at 600 °C. Ngernyen et al. [45] prepared AC from *Eucalyptus* and wattle wood charcoal by CO_2_ and high temperature activation and evaluated the effect of residence time and temperature on the duration of charcoal burning due to the C-CO_2_ gasification reactions. They found that temperature had an evident effect on the extent of burning, particularly at temperatures and times that exceeded 700 °C and 120 min, respectively, after which the burn-off increased rapidly with increasing temperature.

### 3.4. Dehydration Heat by DSC and True Density of the Particles

The differential scanning calorimeter (Co-DSC) curves (Figure 4e) for sawdust, charcoal and AC showed an endotherm peak that was spread broadly between 30 and 150 °C and maximum values near 90 to 100 °C, that was assumed to be due to the loss of adsorbed water (dehydration). The highest latent heat of dehydration of 259 ± 46 J·g^−1^ was evidenced in the AC sample (Table 2). When analyzing the dry AC sample in the same temperature range, the endothermic peak almost disappeared, as shown in Figure 4e. The latent heat of dehydration measured by Co-DSC was due to water vapor adsorbed by the samples [32,33]. Since crystalline species adsorb a small amount of water, the sorption of water in the samples occurred mainly in the porous and amorphous regions. Since the degree of crystallinity in vegetable-derived charcoal samples and AC decreased due to pyrolysis, a high endothermic peak is expected from high water vaporization. The large endotherm is attributed to the large number of OH^−^ groups accessible for water adsorption. Furthermore, the dehydration had peaks occurring at higher temperature in the AC sample than in the charcoal or sawdust samples.

ACs exhibited micro to nanoporous structures and a high specific surface area. The high values indicate large reactivity and adsorption capacity for gases and liquids [46]. Table 2 shows that the true density of AC increased in relation to sawdust or charcoal samples. High pyrolysis temperatures result in bond breaks and condensation reactions in the internal structure of the AC, increasing the true density of graphite-like structures formed from aromatic compounds of lignin. Activation densifies the internal microstructure, producing numerous new micropores that result from the fusion and splitting up of the large pores. The high temperatures of activation (>800 °C) led to more complete cyclic aromatic graphite structures and a large true density [46]. Activating gases react with carbon atoms in the aromatic compounds of lignin to form micropores rather than macro and mesopores, resulting in a large surface area per unit weight [47]. The increase in CO_2_ concentration enhanced the C-CO_2_ reaction that increased the burn-off level, hence pore development in the ACs [45].

### 3.5. Surface Area and Porosity Structure

The textural properties of the AC samples can be seen in Table 3. The isotherm adsorption curve is Type II (Figure 5a) and shows that AC has high N_2_ adsorption at low relative pressures. This type of curve usually indicates non-porous to macroporous material [48], however, our measurements of pore volume indicate that AC has numerous small pores.

The surface area for AC in this study was lower than those reported in most other studies, except for pinewood. In contrast, micropore volume was greater for AC in this study than that found in most of the literature. While micropores in ACs have a high capacity for adsorption of small molecules, such as gases and solvents, AC in this study also had a considerable number of mesopores. The development of mesopores could be associated with the destruction of existing internal walls between micropores due to the high activation temperature (850 °C) and to the anatomical structures (vessels, pores, fibers, pits, etc.) in the native sawdust. Ngernyen et al. [45] verified the formation of mesopores and micropores after evaluating the effect of activation temperature, activation time and CO_2_ concentration on the AC porous properties. They found that higher activation temperatures, longer activation times and higher CO_2_ concentrations led to an increase in BET specific surface area, of micropore volume and total pore volume. However, they showed that temperature had a greater affect.

Mesopores are more desirable than micropores for liquid phase applications, such as adsorption of organic acids [49]. The increase in surface area changes the surface chemistry of AC by providing more binding sites for impurities. AC adsorption is not only influenced by the pore size and surface area, but also by the presence of oxygen groups, particularly with the adsorption of polar molecules [22].

### 3.6. Water Sorption Isotherms

Adsorption-desorption isotherms of water in sawdust, charcoal and AC are shown in Figure 5b–d. Sawdust showed increasing mass with increasing relative humidity. The isotherms also showed hysteresis, which is typical of wood. The shape of the sorption isotherm for sawdust is Type II, sigmoid, and is characteristic of sorption processes with significant heat of sorption and for solution gas–solid of deformable solids [50]. For both sawdust and AC, the desorption curves were higher than the adsorption curves, indicating that the process was irreversible. In comparison, charcoal showed no hysteresis, indicating the process was reversible.

Irreversible water phase change led to hysteresis in porous and swollen materials, leading to irreversible swelling of the adsorbent and capillary condensation-evaporation processes, with water vapor being condensed and confined in the pores [51]. The weakly bound water may form small hysteresis in materials with high swelling and sorption properties. In contrast, large hysteresis for materials with lower swelling and sorption properties might indicate the strong association of carbon chains and bound water [52]. The result of modification of the adsorption/desorption mechanisms may cause changes in the hysteresis curve form, possibly due to the gradual obstruction of the micropore systems in high relative humidity [53,54].

AC had higher moisture equilibrium values than sawdust and charcoal. The higher values were due to the formation of micro-and mesopores in AC, leading to many sites for moisture sorption. Indeed, the first molecules adsorbed at these micropore sites were water, which resulted in the formation of water clusters that eventually grew and formed bridges between pore walls at high relative humidities [55,56,57].

## 4. Conclusions

Activated carbon (AC) from massaranduba sawdust was produced and characterized and compared to ACs from other biomass sources. AC from massaranduba sawdust presented higher reactivity with moisture, lower crystallinity, and higher thermal stability compared to its sawdust and charcoal precursors. FESEM images showed the formation of macro, meso and micropores after the charcoal activation process. DSC was effective in detecting changes in moisture adsorption and heat of dehydration of the different treatments. This study demonstrates the possibility of adding value to massaranduba residues by contributing important information regarding the production and characterization of ACs from wood residues of the Amazon region and providing evidence that the massaranduba sawdust may be used to scale-up the production of ACs.

## Figures and Tables

**Figure 1 polymers-11-01276-f001:**
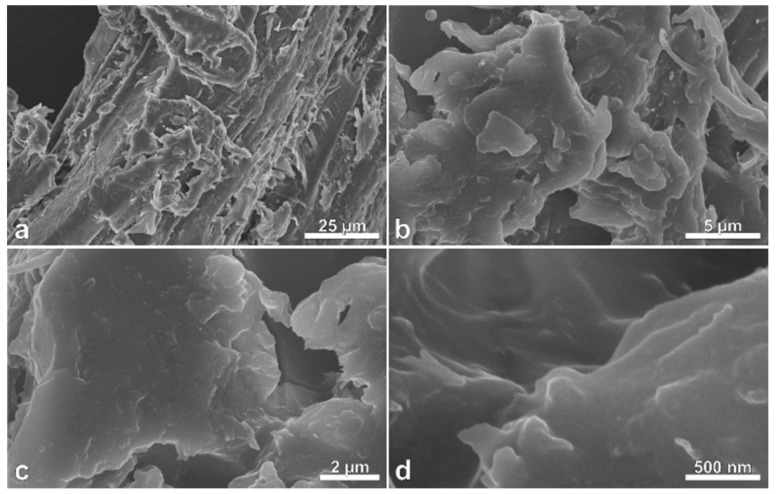
Field emission scanning electron photomicrographs (FESEM) of massaranduba (*Manilkara huberi*) sawdust showing particles surfaces (**a**–**d**), which consist mostly of highly lignified cell wall material.

**Figure 2 polymers-11-01276-f002:**
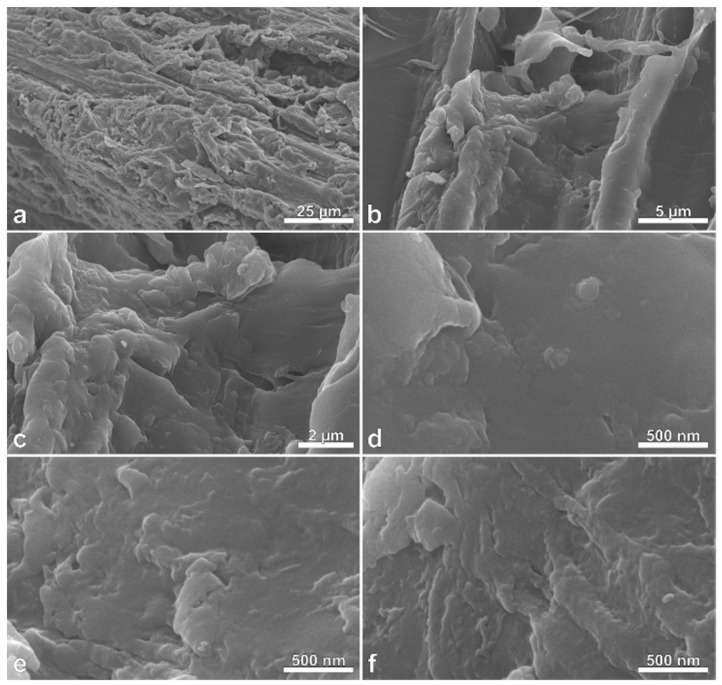
Field emission scanning electron photomicrographs (FESEM) of charcoal produced from massaranduba (*Manilkara huberi*) sawdust at various magnifications, showing the sample surfaces that consist primarily of cell wall material (**a**–**f**). Note the lack of any significant porous structures in the samples.

**Figure 3 polymers-11-01276-f003:**
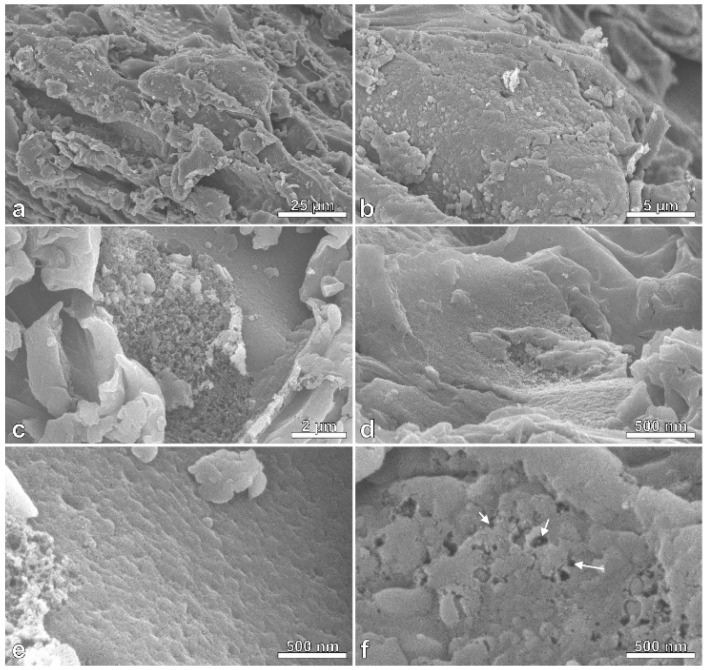
Field emission scanning electron photomicrographs (FESEM) of activated carbon (AC) produced from the charcoal of massaranduba (*Manilkara huberi*) sawdust, showing the particle surfaces (**a**–**f**). Note the porosity that was developed and visible at virtually all magnifications shown, and the appearance of nanoscale pores (**f**, arrows).

**Figure 4 polymers-11-01276-f004:**
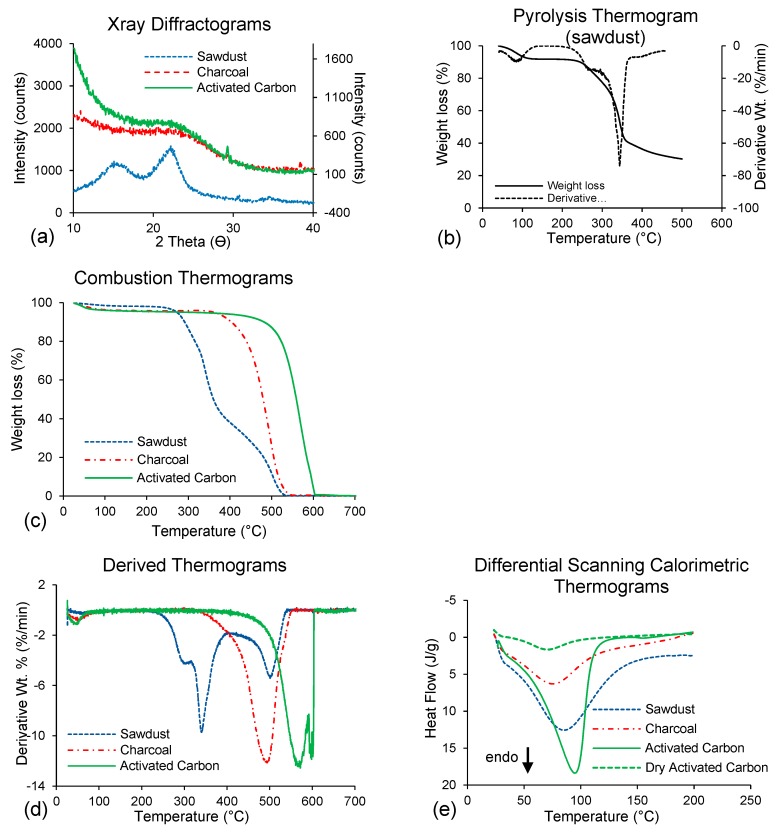
Graphic analyses demonstrating physical properties of massaranduba (*Manilkara huberi*) sawdust and its products (charcoal and activated carbon): (**a**) X-ray diffractograms; (**b**) Thermogravimetric Pyrolysis analysis-Py-TGA (only for sawdust); (**c**) Thermogravimetric Combustion analysis (Co-TGA) (**d**) Derived thermogravimetry of combustion (Co-DTDA); (**e**) Differential Scanning Calorimetry of combustion (Co-DSC).

**Figure 5 polymers-11-01276-f005:**
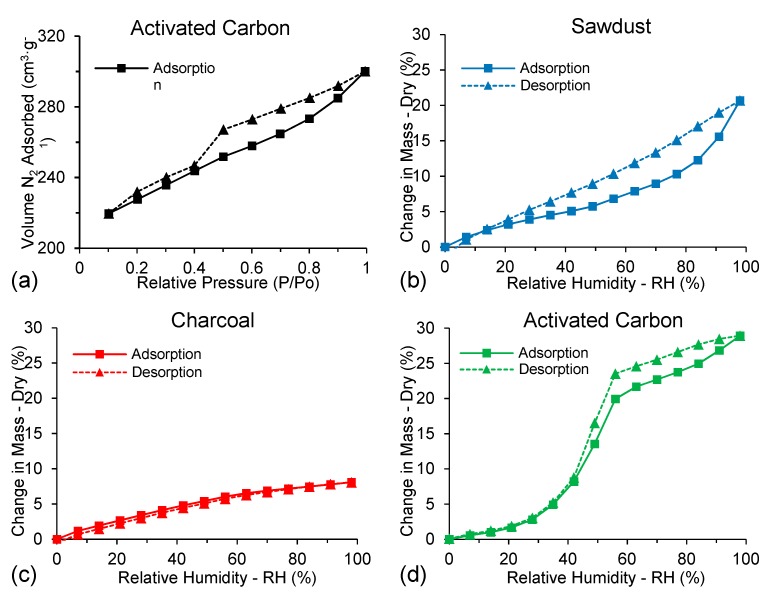
Isotherms of materials derived from massaranduba showing adsorption/desorption of nitrogen (gas) at −196 °C for activated carbon (AC) (**a**); and water sorption/desorption isotherms for sawdust (**b**), charcoal (**c**) and activated carbon (AC) (**d**).

**Table 1 polymers-11-01276-t001:** Summary of activation conditions, surface area characteristics (SBET) and micropore volume of different activated carbons.

Raw Materials	Method and/or Activating Agent Used	Temperature (°C)	S BET ^a^ (m^2^·g^−1^)	Micropore (cm^−3^·g^−1^)	Reference
Pinewood	CO_2_	800	569	NA ^b^	[7]
Guava seeds	CO_2_:H_2_O	850	1201	0.43	[8]
Wood sawdust	ZnCl_2_	500	1301	0.37	[9]
Extractive-free piassava	CO_2_	850	597	0.93	[10]
Olive stones	H_2_O:N_2_	750	807	0.30	[11]
Kenaf	Heat and Vacuum	1100	1742	NA ^b^	[12]

^a^ Brunauer-Emmett-Teller specific surface area. ^b^ Not available.

**Table 2 polymers-11-01276-t002:** Onset degradation temperature (T_onset_) obtained by thermogravimetry (C0-TGA), heat of dehydration determined by differential scanning calorimetry (Co-DSC) and true density determined by gas pycnometry for sawdust, charcoal and AC.

Materials ^a^	T_onset_ (°C)	Heat of Dehydration (J·g^−1^)	True Density (g.cm^−3^)
Sawdust	272	174 ± 19	1.42 ± 0.00
Charcoal	457	90 ± 15	1.44 ± 0.02
Activated carbon (AC)	531	259 ± 46	2.64 ± 0.50

^a^ All materials derived from massaranduba (*Manilkara huberi)* sawdust.

**Table 3 polymers-11-01276-t003:** Textural properties of the massaranduba activated carbon (AC).

Area BET (m^2^·g^−1^)	Volume (m^3^ g^−1^)			
	Macropore	Mesopore	Micropore	Total Pore
698	0.048	0.317	0.621	0.986

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
