# Peer review of "Massaranduba Sawdust: A Potential Source of Charcoal and Activated Carbon"

_polymers, 2019, doi:10.3390/polym11081276_

Round 1

Reviewer 1 Report

The authors reported the utilization of Massarunduba sawdust to produce not only charcoal but also activated carbon, aimed to obtain added value products from waste materials. Therefore, they prepared charcoal by means of a pyrolysis procedure, and then activating that charcoal in order to generate activated carbon. However, prior further consideration for publication in the current form, I have some major concerns about this manuscript.

Comment 1. My main concern is about the possible lack of novelty of the present study. The experimental work seems to be fair, but the authors do not clarify which is the novelty, compared to the uncountable number of studies dealing with the production of charcoal or activated carbon from different wooden materials. That appears to have been already deeply evaluated.

Comment 2. The written English must be considerably improved, since throughout the whole manuscript I found many grammatical and spelling errors (Lines 21, 36, 137, etc.). Among other examples, in lines 274 and 301 the authors wrote “CA sample…” and, probably the meant “AC sample…”. Moreover, along the body of the manuscript, a lack coherence and cohesiveness are clearly observed. The authors are recommended to reread the manuscript and to improve the language of the manuscript.

Comment 3. The authors are encouraged to improve the introduction section. So far, it does not suitably establish the background leading to the proposed investigation.

Comment 4. Some data are missed to support the statement appearing in the introduction. For instance, in Lines 85-87 it is required to include the wood production of the waste generated during the industrial exploitation of wooden materials coming from the Amazon. Thus, it would be much clearer the need for the implementation of processes to obtain such added value materials.

Comment 5. The authors must use a coherent quoting methodology (Line 39).

Comment 6. More information about the raw material is required such as purity, composition, the company name/provenance, etc.

Comment 7. (Lines 110-115) Why did the authors change the activation procedure from that cited? Did the authors optimise the activation process? In that case, it should be included in this manuscript. They should have first optimised the activation procedure to clarify why they applied such conditions.

Comment 8. Figures 4 and 5 should be reinserted since they are not completely seen. Graphs (a) and (b) seem to be overlapped those found below them (c) and (d) in both cases (Lines 208 and 318). On the other hand, Figure 4 (d) should present an arrow indicating the direction in which the thermal events are endothermic, as usual in this type of plots.

Comment 9. (Lines 210-225). It is required to reduce the Figure 4 caption up to a two-line sentence. More than that is quite inappropriate. The results explanation or analysis must be included in the body of the text.

Comment 10. Did the authors apply the TGA tests over the sawdust under inert atmosphere to corroborate that at 500 ÂşC its pyrolysis takes place? This fact should be confirmed, given that they have the required equipment to do that.

Comment 11. The authors should check all the inconsistencies related to the temperature values which do not correspond what can be seen in the graphs, such as the maximum degradation temperatures of charcoal and activated carbon in Line 221, since those values differ with both the maximum decomposition rate and the ultimate decomposition temperature. Similar disparities were found along the result section (Lines 249, 251, 261, etc.).

Comment 12. (Lines 229-233). It is hard to compare the results obtained under Nitrogen atmosphere with those obtained by these authors when applying the TGA tests using air, even more importantly when they did not apply such sort of TGA tests at inert conditions.

Comment 13. In lines 240-243 the authors analyse the decomposition pattern of sawdust, considering that some pyrolysis occurs. However, under air condition not any “pyrolysis process” can be observed. The authors are encouraged to rewrite this part, just in case they meant something different.

Comment 14. The authors cannot affirm that “the dehydration had peaks occurring at higher temperatures for pyrolyzed samples (charcoal and AC)” (Lines 282-283) since that found for charcoal appears at a lower temperature than the corresponding to the sawdust used as raw material.

Comment 15. It is not so clear which is the conclusion that can be drawn from the water sorption isotherm tests. That section should be rewritten somehow to elucidate the main conclusions of these results. The only fact clear here is that the sorption process is irreversible for the starting sawdust and the activated carbon, while it is not in the case of the charcoal.

Author Response

Thank you for your useful comments and suggestions on the structure of our manuscript. We have modified the manuscript accordingly with suggested and the detailed corrections are attached:

Reviewer 2 Report

This manuscript reports the pyrolytic conversion of saw dust into charcoal and the posterior activation with CO2 to render activated carbon. The final material displays micro/mesoporosity and sizable specific surface area and pore volume well in the range of what can be expected from lignocellulosic biomass with the employed method. The array of characterization techniques gives a consistent picture of the material and its transformation. The used biomass seems to be of relevance in South America and therefore, an addition of value to this residue is of interest. I recommend publication after minor revision.

-        Could the authors indicate the chemical composition of this residue? The relation of cellulose to lignin and hemicellulose. Especially the ratio cellulose:lignin can be a determinant for the final carbon properties. See therefore DOI: 10.1002/ente.201800685. Could you please comment.

-           The knowledge of the composition is also crucial for the unambiguous interpretation of the thermal analysis results. Again, see DOI: 10.1002/ente.201800685 and for instance DOI: 10.1021/acssuschemeng.6b00388. In fact, to discern the decomposition of the individual lignocellulose components I strongly advice to perform TG under nitrogen atmosphere and not air as in the present case. Importantly, the second weight loss of sawdust (fig 4b) can be attributed to oxidative degradation of char formed during the first weight loss and not only on the degradation of lignin as suggested in the manuscript.  Please revise and comment.

-           Line 274: Should “CA” read “AC”? There are several other typos and language issues throughout the manuscript. Please revise. For instance line 280: ”it is expected an endothermic peak higher water vaporization”

-           Textural properties page 9: I find it hard to follow the explanation of the increased density due to CO2 activation (section l287-299). This section is a bit confusing and should be revised, possibly restructured to make the point clearer. I also don´t understand why creating microporosity increases the true density?

-           Line 313, 353: incomplete sentences

Author Response

Thank you for your useful comments and suggestions on the structure of our manuscript. We have modified the manuscript accordingly with suggested and the detailed corrections are attached.

Round 2

Reviewer 1 Report

I include in attachment my arguments to the atuhor responses. Although they have improved their manuscript, there are still some basic issues to be solved.

Author Response

Thanks again for the useful comments on the manuscript. I hope that we have answered all of the concerns of the reviewer. Our responses are all in green.

Round 3

Reviewer 1 Report

I send in attachment my responses to the author's reply (in violet).

Author Response

Dear Reviewer 1,

I think we have successfully addressed your concerns on the manuscript and do thank you very much for your excellent and timely review!

Please let me know if you require any more modifications. My responses are in orange on the attached. All the boxes are checked as "can be improved" but I'm not really sure what to improve.

Best regards,

Delilah Wood
